**Subject Category:**
Biology (whole organism)

health and disease and epidemiology/computational biology

avian influenza, dynamic model, bird migration, virus translocation

**Author for correspondence:**
Jeffrey Shaman
e-mail: jls106@columbia.edu

# Pathobiological features favouring the intercontinental dissemination of highly pathogenic avian influenza virus

## Xueying Li[1,2], Bing Xu[1,3] and Jeffrey Shaman[2]

[1]Department of Earth System Science, Ministry of Education Key Laboratory for Earth System Modelling, Tsinghua, Beijing, People's Republic of China
[2]Department of Environmental Health Sciences, Columbia University, New York, NY, USA
[3]State Key Laboratory of Remote Sensing Science, College of Global Change and Earth System Science, Beijing Normal University, Beijing, People's Republic of China

XL, 0000-0003-1307-9187; JS, 0000-0002-7216-7809

Avian influenza viruses (AIVs) are a continued threat to global health and economy. Unlike other highly pathogenic AIVs, novel H5N8 disseminated very quickly from Korea to other areas in Asia, Europe and even North America following its first outbreak in 2014. However, the pathobiological features of the virus that favoured its global translocation remain unknown. In this study, we used a compartmental model to examine the avian epidemiological characteristics that would support the geographical spread of influenza by bird migration, and to provide recommendations for AIV surveillance in wild bird populations. We simulated virus transmission and translocation in a migratory bird population while varying four system properties: (i) contact transmission rate; (ii) infection recovery rate; (iii) mortality rate induced by infection; and (iv) migratory recovery rate. Using these simulations, we then calculated extinction and translocation probabilities for influenza during spring migration as a function of the altered properties. We find that lower infection recovery rates increase the likelihood of AIV translocation in migratory bird populations. In addition, lower mortality rates or migration recovery rates also favour translocation. Our results identify pathobiological features supporting AIV intercontinental dissemination risk and suggest that characteristic differences exist among H5N8 and other AIV subtypes that have not translocated as rapidly (e.g. H5N6 and H5N1).

# 1. Introduction

During the twentieth century, type A influenza viruses caused three massive global pandemics: 1918 (H1N1), 1957 (H2N2) and 1968 (H3N2) [1–3]. A fourth type A influenza pandemic took place in 2009 [4]. Wild birds are the natural host of type A influenza [5] and have seeded multiple influenza spillover events. In 1997, an outbreak of avian influenza virus (AIV) subtype H5N1 emerged in Hong Kong [6]. This highly pathogenic avian influenza virus (HPAIV) caused a large-scale outbreak globally and has become endemic in some regions [7]. More than 140 million domestic birds have been killed by the virus or culled to control the virus [8]. Additionally, hundreds of humans have been infected with H5N1 after direct contact with poultry, and the case fatality rate in infected humans is over 50% [9,10].

Since the emergence of H5N1, many other influenza viruses have been isolated from birds. During 1999 and 2003, respectively, subtypes H7N1 and H7N3 were documented in Italy [11], and more recently, H7N9, H10N8, H5N8 and H5N6 have emerged and presented challenges to both avian and human health [12–15]. Low pathogenic avian influenza virus (LPAIV) H7N9, in particular, has infected more than 1500 humans in China since 2013 with a human case fatality rate of 41% [9,16].

AIVs are now regularly isolated in wild birds throughout the world. Due to outbreaks in humans and consequent bird culling, AIV remains a critical economic threat to the global poultry industry. Further, there is concern that a full-fledged human pandemic with a high case fatality rate might occur, should one of these subtypes mutate or reassort and gain the capacity for efficient human-to-human transmission [17,18].

Previous research has shown that bird migration is an important means of dissemination for AIV. Within the wild bird reservoir, type A influenza viruses persist in evolutionary equilibrium. Some highly pathogenic AIVs only cause mild symptoms in anseriformes species, which allows these birds to spread HPAIV globally during migration [19]. Overlap among habitats of different waterfowl species forms a bridge of intercontinental dissemination [20]. Previous research indicates that the timing of HPAIV H5N1 outbreaks corresponds with the timing of bird migration [21]. In addition, observed associations between viral evolution flow and bird migration in Asia have provided further evidence of the importance of bird migration for virus dissemination. Wild bird movement may also seed infection among other species at different habitats along migration routes [22], which may accelerate virus evolution or reassortment and further increase the risk of an outbreak or pandemic [23].

Unfortunately, surveillance of AIV in wild bird populations remains limited. Most wild bird sampling is confined to dead birds and typically does not provide timely warning for preventive actions. Information that can help guide more effective surveillance is urgently needed.

HPAIV H5N8 virus was first isolated in China in 2010. This virus originated from the H5N1 Gs/GD lineage [24]. Since 2014, H5N8 from Korea has formed a large clade on the phylogenetic tree that consists of viruses sampled from Japan, Taiwan, Europe and North America [25–31]. These viruses have caused large-scale outbreaks and subsequently reassorted with local AIV.

Unlike H5N8, which translocated from Asia to North America by bird migration through Beringia [32], many other avian influenza subtypes have remained geographically isolated and evolved separately in the Eurasian and American areas [20]. For instance, wild bird migration helped to spread H5N1 virus in Eurasia, but geographical isolation to date has been sufficient to stop dissemination of the Gs/GD lineage to the American continents [33]. This bottleneck can also be seen in the spread of H5N6, which has been isolated in East Asia from 2014 to 2017 and only spread to Europe in the winter of 2017–2018 [34].

It remains unclear why H5N8 virus has been able to spread faster and farther by bird migration than either H5N6 or H5N1. In this paper, we use a compartmental model to evaluate how far an AIV can travel in a migratory bird population given varying epidemiological properties. We simulate virus transmission and translocation in a migratory bird population while varying four system properties: (i) contact transmission rate; (ii) infection recovery rate; (iii) mortality rate induced by infection; and (iv) migratory recovery rate. The objectives are to determine which features support the geographical spread of influenza by bird migration and to provide recommendations on surveillance of avian influenza in wild bird populations.

# 2. Methods

## 2.1. The model

We developed a model based on the form presented in [35]. The model simulates the transmission and translocation of AIV in a typical mallard population (around 5000 individuals). The model represents

eight distinct geographical patches along a migration route, one patch for the wintering ground (patch 1), three for stopover during spring migration (patches 2–4), one for the summer breeding ground (patch 5) and three for stopover during return autumn migration (patches 6–8). The annual migration cycle is presented in figure 1.

The model consists of five equations (equation (2.1)) for each patch $i = 1, 2, \ldots, 8$. Specifically, we have

$$\frac{dS_i(t)}{dt} = -\beta_i S_i(t)[I_{1,i}(t) + I_{2,i}(t)] - (m_{i,t} + \mu_n + \mu_{h,i})S_i(t) + m_{i-1,t}S_{i-1}(t) + b_{i,t}, \tag{2.1a}$$

$$\frac{dI_{1,i}(t)}{dt} = \beta_i S_i(t)[I_{1,i}(t) + I_{2,i}(t)] - (v + r + \mu_d + \mu_n + \mu_{h,i})I_{1,i}(t), \tag{2.1b}$$

$$\frac{dI_{2,i}(t)}{dt} = vI_{1,i}(t) - (r + m_{i,t} + \mu_n + \mu_{h,i})I_{2,i}(t) + m_{i-1,t}I_{2,i-1}(t), \tag{2.1c}$$

$$\frac{dR_{1,i}(t)}{dt} = rI_{1,i}(t) - (v + \mu_n + \mu_{h,i})R_{1,i}(t) \tag{2.1d}$$

and

$$\frac{dR_{2,i}(t)}{dt} = vR_{1,i}(t) + rI_{2,i}(t) - (m_{i,t} + \mu_n + \mu_{h,i})R_{2,i}(t) + m_{i-1,t}R_{2,i-1}(t). \tag{2.1e}$$

Model parameters are defined in table 1, and model state variables are defined in table 2. For each patch, virus transmission is simulated using an SIR type model. We consider a migration delay induced by infection in the model, so the infected (I) and recovered (R) compartments are divided into $I_1$, $I_2$ and $R_1$, $R_2$, respectively. Birds in the $I_1$ and $R_1$ compartments are not healthy enough to migrate but can move to $I_2$ and $R_2$ with a migration recovery rate ($v$). Birds in $I_1$ and $I_2$ can recover to $R_1$ and $R_2$, respectively, with an infection recovery rate ($r$). The migration recovery rate and the infection recovery rate are independent of each other. We include a mortality rate induced by infection in the $I_1$ compartment. We do not consider cross-immunity and immunity loss in the model.

Birds remain at each stopover site for at least 20 days, at the wintering ground for at least 91 days and at the breeding ground for at least 153 days. The average time of migration between neighbouring patches is 1 day. A migration rate matrix is used to simulate movement. Provided a bird is healthy enough to migrate, the rate of leaving patch $i$ at time $t$ is $m_{i,t} = 0$ if the birds are scheduled to be at patch $i$ on day $t$, and $m_{i,t} = 1$ if the birds are not scheduled to be at patch $i$ on day $t$. Birth occurs beginning six weeks post-arrival at the breeding ground, continues for one month and occurs at a fixed rate of 40 new birds per day. A natural mortality rate $\mu_n$ is included in all patches and a hunting mortality rate $\mu_h$ is represented in patch 1 (wintering ground) and patches 6–8 (autumn migration stopover sites) (table 1).

Previous studies have shown that environmental transmission is important for the transmission and persistence of AIV in a particular location. Breban et al. [40] and Rohani et al. [41] added an environmental reservoir into SIR models and showed that the environmental transmission rate— which is, on average, hundreds of times smaller than the contact transmission rate—helps LPAIV persist in the environment and infect additional susceptible hosts, and can also generate secondary outbreaks. However, it is unclear whether environmental transmission supports the translocation of AIV. We therefore tested a second version of the model that includes environmental transmission. For each patch $i = 1, 2, \ldots, 8$, we have

$$\frac{dS_i(t)}{dt} = -\beta_i S_i(t)[I_{1,i}(t) + I_{2,i}(t)] - \gamma(1 - e^{-\alpha E_i(t)})S_i(t) - (m_{i,t} + \mu_n + \mu_{h,i})S_i(t) + m_{i-1,t}S_{i-1}(t) + b_{i,t}, \tag{2.2a}$$

$$\frac{dI_{1,i}(t)}{dt} = \beta_i S_i(t)[I_{1,i}(t) + I_{2,i}(t)] + \gamma(1 - e^{-\alpha E_i(t)})S_i(t) - (v + r + \mu_d + \mu_n + \mu_{h,i})I_{1,i}(t), \tag{2.2b}$$

$$\frac{dI_{2,i}(t)}{dt} = vI_{1,i}(t) - (r + m_{i,t} + \mu_n + \mu_{h,i})I_{2,i}(t) + m_{i-1,t}I_{2,i-1}(t), \tag{2.2c}$$

$$\frac{dR_{1,i}(t)}{dt} = rI_{1,i}(t) - (v + \mu_n + \mu_{h,i})R_{1,i}(t), \tag{2.2d}$$

$$\frac{dR_{2,i}(t)}{dt} = vR_{1,i}(t) + rI_{2,i}(t) - (m_{i,t} + \mu_n + \mu_{h,i})R_{2,i}(t) + m_{i-1,t}R_{2,i-1}(t) \tag{2.2e}$$

and

$$\frac{dE_i(t)}{dt} = I_{1,i}(t) + I_{2,i}(t) - cE_i(t). \tag{2.2f}$$

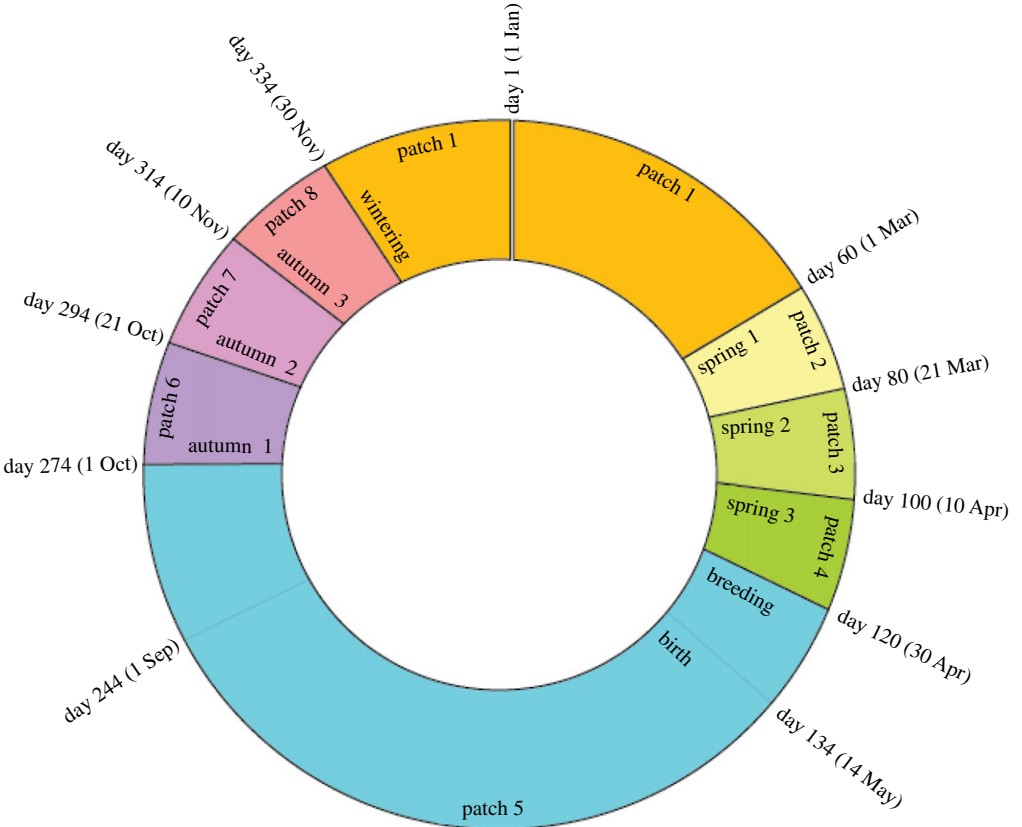

**Figure 1.** Annual migration schedule of the simulated migratory bird population. The model starts on day 1 and ends on day 365 of a year. The wintering season is from day 334 of the preceding year to day 60 of the next year (patch 1).

Infected birds shed virus into the environment where the virus can persist. In this form, the virion population in the environment, $V$, is affected by two processes

$$\frac{dV}{dt} = \omega I - \eta V, \tag{2.3}$$

where $I$ is the number of infected individuals, $\omega$ is the virus shedding rate and $\eta$ is the virus decay rate in the environment. Dividing equation (2.3) by $\omega$ yields equation (2.2f), where $E = V/\omega$ and $c = \eta$.

We assume that, at each patch, infection occurs through both contact transmission and environmental transmission. The environmental transmission per susceptible bird is derived from Breban *et al.* [40] and is given by $\gamma(1 - e^{-\sigma V})$. A constant exposure rate, $\gamma$, was introduced to represent the virus consumption rate scaled by lake volume. The parameter, $\sigma$, is a constant rate related to the empirically determined $ID_{50}$. After dividing equation (2.3) by $\omega$, the environmental transmission rate becomes $\gamma(1 - e^{-\alpha E})$. We define $\alpha = \sigma\omega$, which is rescaled environmental infectiousness.

## 2.2. Simulations and analysis

The aim of this study is to determine which epidemiological features support the migratory dispersal of influenza from the wintering grounds. To simulate differences in epidemiological features, we first used the baseline equation (2.1) model and varied four model parameters: the contact transmission rate ($\beta$), the recovery rate ($r$), the mortality rate induced by infection ($\mu_d$) and the migration recovery rate ($v$). For contact transmission rate, we followed the assumptions of Galsworthy *et al.* [42] that the contact rate is density-dependent and the transmission rate at the summer breeding ground is 25% of the rate at other times of the year. We then varied the transmission rate from $\beta = 0.2 \times 10^{-4}$ to $\beta = 3 \times 10^{-4}$ (bird$^{-1}$day$^{-1}$). The recovery rate is calculated as the reciprocal of infection period (days) and ranges from $r = 1/3$ to $r = 1/3$ (days$^{-1}$), based on previous modelling and virus experiments [42,43]. We also varied the infection-induced mortality rate from $\mu_d = 0$ to $\mu_d = 1$ (days$^{-1}$) and the migration recovery rate from $v = 1/100$ to $v = 1/1$ (days$^{-1}$). All parameters' ranges are shown in table 1. In a second suite of simulations using the model with environmental transmission (equation (2.2)), we varied the same four baseline model parameters ($\beta$, $r$, $\mu_d$, $v$) as well as $c$, $\gamma$ and $\alpha$ (see table 1 for ranges).

**Table 1.** The parameters in the model.

| parameter | description | value | unit | reference |
|---|---|---|---|---|
| $\beta_i$ | contact transmission rate at patch $i$ | for $i \neq 5$, $\beta_i = b = [0.2 \times 10^{-4}, 3 \times 10^{-4}]$; for $i = 5$, $\beta_i = b/4$ | $\text{bird}^{-1}\,\text{day}^{-1}$ | [35,36] |
| $r$ | infection recovery rate | $r = [1/13, 1/3]$ | $\text{day}^{-1}$ | [35,37] |
| $\mu_d$ | mortality rate induced by infection | $\mu_d = [0, 1]$ | $\text{day}^{-1}$ | |
| $v$ | migration recovery rate | $v = [1/100, 1/1]$ | $\text{day}^{-1}$ | [35,38] |
| $\mu_n$ | natural mortality rate | $\mu_n = 0.315/365$ | $\text{day}^{-1}$ | [35] |
| $\mu_{h,i}$ | hunting mortality rate at patch $i$ | for $i = 5$, $\mu_{h,i} = 0.320/365$ | $\text{day}^{-1}$ | [35,39] |
| $m_{i,t}$ | migration rate | shown in figure 1 | $\text{day}^{-1}$ | |
| $b_{i,t}$ | birth rate | for $i = 5$ and $162 < t < 216$, $b_{i,t} = 40$, otherwise $b_{i,t} = 0$ | $\text{birds day}^{-1}$ | [35] |
| $c$ | persistence of virions | $c = [2/365, 20/365]$ | $\text{day}^{-1}$ | [40] |
| $\gamma$ | exposure rate | $\gamma = [1 \times 10^{-3}/365, 5 \times 10^{-3}/365]$ | $\text{day}^{-1}$ | [40] |
| $\alpha$ | rescaled environmental infectiousness | $\alpha = [1/365, 1 \times 10^{6}/365]$ | $\text{bird}^{-1}\,\text{day}^{-1}$ | [40] |

**Table 2.** The variables of the model.

| variable | definition | initial value |
|---|---|---|
| $S$ | susceptible birds | $S(0) = 4990$ |
| $I_1$ | infected birds which are not healthy enough to migrate | $I_1(0) = 10$ |
| $I_2$ | infected birds with migration ability | $I_2(0) = 0$ |
| $R_1$ | recovered birds which are not healthy enough to migrate | $R_1(0) = 0$ |
| $R_2$ | recovered birds with migration ability | $R_2(0) = 0$ |
| $E$ | virions in the environment divided by virus shedding rate ($V/\omega$) | $E(0) = 1$ |

Both models were implemented in R (v. 3.5.2) and integrated with a time step of 0.2 days for 365 days. We used Latin hypercube sampling (LHS) to sample different parameter values based on table 1 ranges. We randomly chose 600 samples for simulation with the baseline model (equation (2.1)) and 1000 samples for the second version with environmental transmission (equation (2.2)). All simulations were run stochastically [44]. Influenza virus was seeded into the simulation on the first day, and the initial model states can be found in table 2. To account for stochastic effects, 100 simulations with each combination of parameters were run.

In these stochastic simulations, the virus can go extinct within the bird population. To measure the likelihood of virus survival and translocation, for each patch and combination of parameters, we calculated the fraction of runs for which the virus is present in the bird population at a given patch, as well as extinct. We then examined all the simulations to determine the combination of pathobiological features that favour virus persistence and geographical dispersal, or, conversely, virus extinction and geographical isolation.

## 3. Results

Patch 5, the summer breeding ground, is the critical site for intercontinental dissemination of virus, as it is where bird populations from different regions would potentially mingle. We therefore focus on the

presence and extinction of virus during spring migration (patches 1–5). Viruses with parameter combinations supporting a greater likelihood of reaching patch 5 are more likely to spread intercontinentally.

Figure 2 shows the marginal likelihood that infection reaches patches 3 and 5 for each of the four parameters of the baseline model. Each point in figure 2 represents the result calculated for a single parameter combination. For each parameter combination, the likelihood of reaching patch 3 or patch 5 is quantified as the fraction of the 100 stochastic simulations for which the virus does not go extinct before reaching the patch. We find that for all parameters, infection is more likely to reach patch 3 than patch 5. A strong sensitivity to infection recovery rate is evident. There is also sensitivity to transmission rate with the probability of reaching patches 3 and 5 rising as transmission rate increases. Patterns can also be seen for infection-induced mortality rate and migration recovery rate. A lower mortality rate increases the probability of reaching the summer grounds, as does a higher migration recovery rate.

Among the four parameters, infection recovery rate exhibits the strongest effect. With an infection period of 13 days (infection recovery rate of $1/13 \, \text{day}^{-1}$), virus reaches patch 3 for the majority of simulations, whereas for an infection period of 3 days (an infection recovery rate of $1/3 \, \text{day}^{-1}$), the majority of distinct parameter combinations have a 0.25 or less likelihood of reaching patch 3. A similar but stronger distinction is seen for patch 5. For almost all parameter combinations with an infection period of 3 days (an infection recovery rate of $1/3 \, \text{day}^{-1}$), the virus never reaches patch 5, whereas with a longer infectious period (a lower infection recovery rate), there is a high likelihood of AIV reaching patch 5. The marginal distribution of extinction exhibits a perfect symmetric pattern with figure 2 (not shown).

To quantify the sensitivity of virus translocation to each parameter, we computed partial rank correlation coefficients (PRCC) based on the result shown in figure 2. A larger absolute value of PRCC demonstrates a stronger correlation between the given parameter and the likelihood of translocating to a given patch. The values of PRCC are shown in table 3. We find that the absolute values of PRCC for transmission rate, infection recovery rate and mortality rate are all 0.6 or higher, indicating that these three parameters strongly affect virus translocation. For translocation to patch 5, infection recovery rate has the largest absolute value of PRCC, indicating the strongest impact. Although the PRCC for migration recovery rate is less than for the other three parameters, this parameter also has a significant impact on the likelihood of virus translocation.

To further investigate the features that always support translocation to a given patch versus never supporting translocation during spring migration, we plotted density histograms for each of the four parameters (figure 3). Always supporting translocation occurs when 100% of the 100 stochastic runs with a particular parameter combination translocate virus to a given patch, and never supporting translocation occurs when 0% of the stochastic runs with a particular parameter combination translocate virus to a given patch. Unlike figure 2, the marginal density histograms of 100% versus 0% translocation likelihood are asymmetric. For example, the likelihood of always translocating with a given combination of parameters is relatively insensitive to the transmission rate parameter (i.e. the histogram is flat). By contrast, the likelihood of never translocating decreases as the transmission rate parameter increases.

There is an increasing likelihood of always translocating to patches 3 and 5 as mortality rates decrease; further, there is an increasing likelihood of virus never translocating as mortality rates increase. For migration recovery rate, with a higher migration recovery rate, virus is more likely to always translocate to patch 5. The strongest effect manifests for the infection recovery rate parameter. Certain arrival of the virus at the summer breeding grounds virtually requires a lower recovery rate; approximately 80% of parameter combinations that always translocated to patch 5 had an infectious period longer than 11 days (infection recovery rate smaller than $1/11 \, \text{day}^{-1}$). For parameter combinations that never translocated to patch 5, the relationship is reversed but weaker.

The marginal likelihoods based on the results using the second model with environmental transmission are shown in figure 4. As for figure 2, each point in figure 4 represents the result calculated for one parameter combination. We again used PRCC to quantify the sensitivity of the likelihood of virus translocation in the second model with environmental transmission, here for the four core parameters and three additional environmental transmission parameters. The findings are very similar to the results generated with the baseline model. A strong sensitivity to infection recovery rate remains with the probability of reaching patch 3 and patch 5 increasing as the infection recovery rate decreases (infection period increases). For contact transmission rate, mortality rate induced by infection, and migration recovery rate, the pattern is also similar to that of the baseline model (figure 2). By contrast, the likelihood of translocation exhibited little sensitivity to the environmental

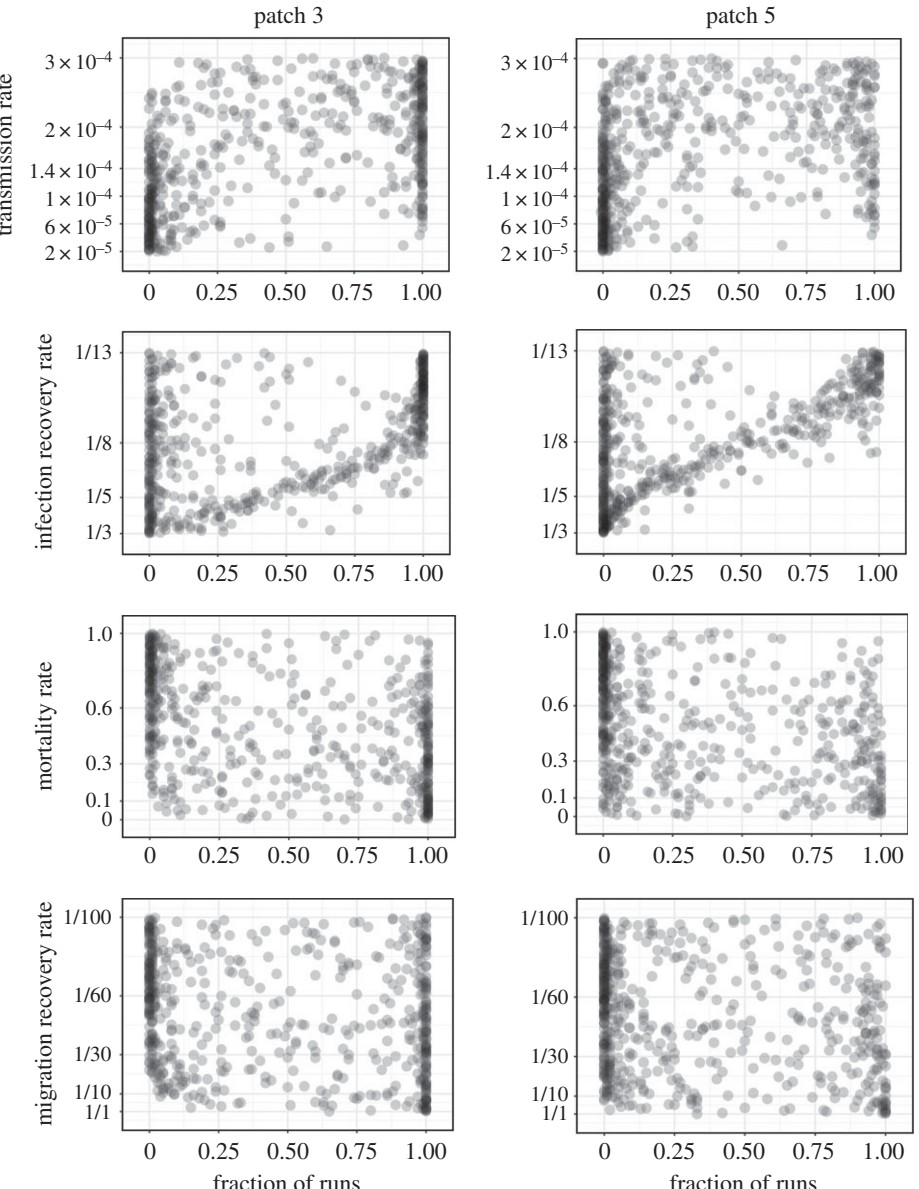

**Figure 2.** Marginal distribution of the likelihood of virus reaching patches 3 and 5 for each of the four varied parameters: transmission rate, infection recovery rate, mortality rate induced by infection and migration recovery rate. Every point in the figure represents the result calculated for one sampled parameter combination. For each parameter combination, the likelihood of reaching patch 3 or patch 5 is quantified as the fraction of runs (of 100 stochastic simulations) for which the virus does not go extinct before reaching that patch.

**Table 3.** PRCC between the likelihood of virus translocating to patch 3 or patch 5 and each varying parameter based on the results of the baseline model.

| input | patch 3 | | patch 5 | |
|---|---|---|---|---|
| parameter | PRCC | *p*-value | PRCC | *p*-value |
| $\beta$ | 0.6947 | 0.00 | 0.6330 | 0.00 |
| $r$ | −0.6144 | 0.00 | −0.6803 | 0.00 |
| $\mu_d$ | −0.6039 | 0.00 | −0.5616 | 0.00 |
| $v$ | 0.3919 | 0.00 | 0.3893 | 0.00 |

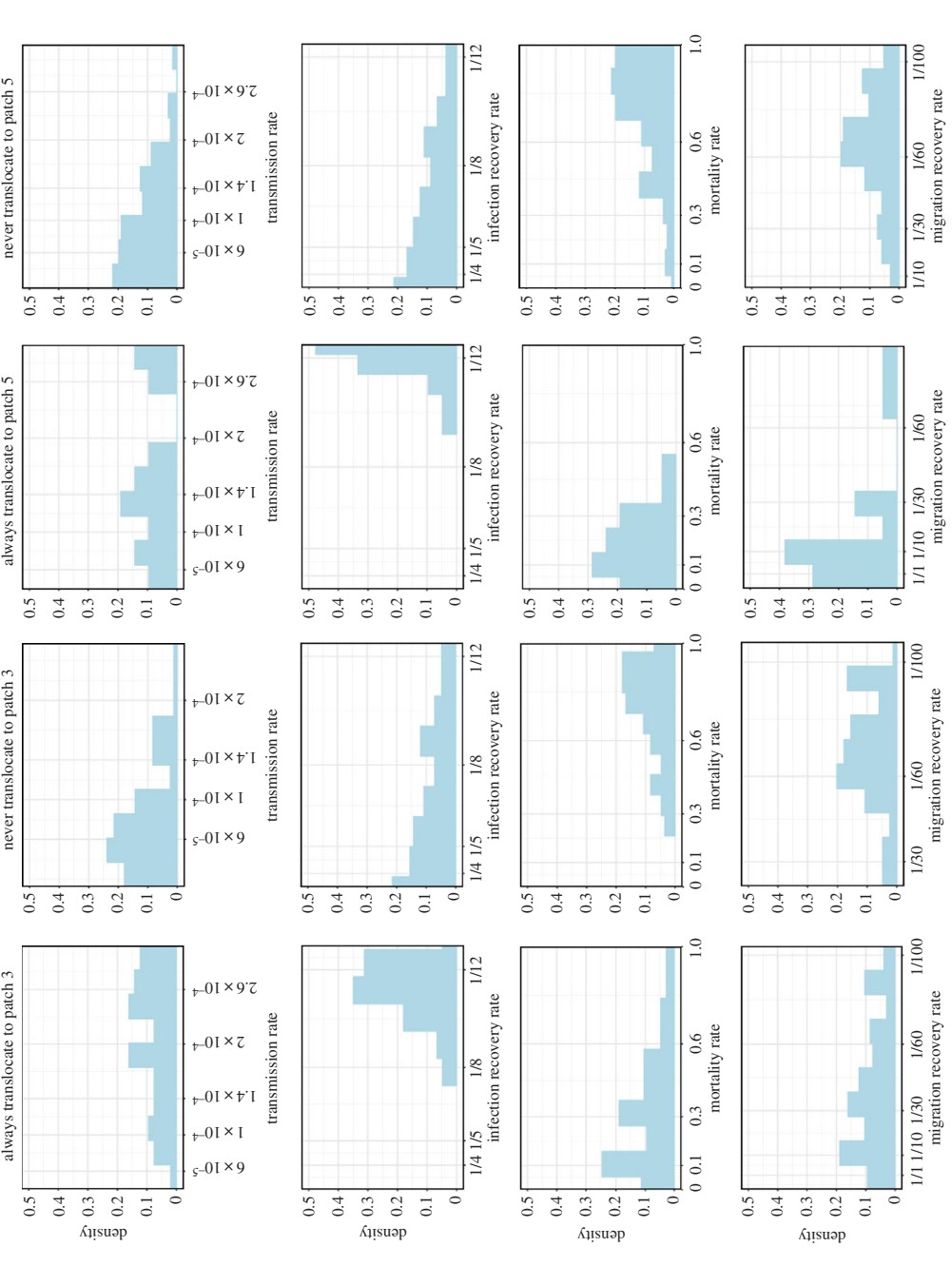

**Figure 3.** Marginal density histograms of virus always translocating and never translocating to patches 3 and 5 plotted as a function of each of the four studied parameters. A parameter combination always supports translocation if the virus appears in a given patch during 100% of runs with that parameter combination and never translocates if the virus appears in a given patch during 0% of runs with that parameter combination. The shown marginal density histograms isolate each parameter in turn and show whether certain values strongly favour or inhibit translocation.

transmission parameters ($c$, $\gamma$, $\alpha$); in particular, the PRCCs are not significant and indicate little correlation between the likelihood of virus translocation and changes to the environmental transmission parameters (table 4). These results show that inclusion of environmental transmission processes does not alter the sensitivity of AIV translocation to the baseline model parameters.

# 4. Discussion

In this paper, we used a compartmental model to simulate the translocation of AIV via a migratory bird population. The model simulates bird movement and virus translocation along bird flyways, which are critical for AIV dissemination. Compared to a simpler SIR model, this model not only represents three bird infection states—susceptible, infected and recovered—but also considers loss and recovery of the ability to migrate induced by infection. These latter conditions can impact the dynamic of infection translocation along the entire flyway. According to Galsworthy *et al.* [35], by including migration delay, the scale of an outbreak becomes smaller than in a simpler SIR model without migration delay, due to the isolation of infected birds from the main population of susceptible birds. Additionally, an AIV with low transmissibility may persist and spread over more locations along the flyway.

Our results indicate that infection recovery rate strongly affects the likelihood of virus dispersal from winter habitats to summer breeding grounds. Transmission rate, mortality rate induced by infection and migration recovery rate also have some influence on virus translocation. A lower infection recovery rate leads to a higher probability of virus translocation to the summer breeding grounds, where the likelihood of intercontinental dissemination would be increased. An avian influenza producing a lower infection recovery rate, i.e. a longer infectious period, has more opportunities for transmission. Conversely, with a higher infection recovery rate, fewer susceptible birds are infected due to the shorter infectious period, which increases the likelihood of virus extinction within a migrating flock.

The results presented here are consistent with observations and analyses of the dispersal of different subtypes of AIV. Novel reassorted HPAI H5N8 virus dispersed quickly to both Europe and North America after its initial outbreak in Korea during 2014 [45]. Animal experiments have shown that the shedding period of H5N8 in ducks ranges from 7 to 13 days, and H5N8 was found to be not fatal to ducks [46–49] and, specifically, asymptomatic in mallard ducks [43,50].

Unlike H5N8, other HPAI viruses have disseminated more slowly and remained geographically isolated. HPAI H5N1 virus was first isolated in wild birds in China during 1996 and became endemic in domestic birds in mainland China. Beginning in 2003, the virus appeared in Europe and Africa [20]; however, the infection rate of H5N1 in Alaska is remarkably low (0.06%), and the virus does not appear elsewhere in the Americas [33]. HPAI H5N1 has an infectious period of around 5 days in ducks, which is shorter than H5N8, and a mortality rate higher than 60% [37]. Based on our model findings, these pathobiological features are less favourable for virus translocation via migratory birds than the features reported for H5N8.

Similarly, HPAI H5N6 has also dispersed more slowly than H5N8. It spread from China to other parts of East Asia during 2014 to 2017, and was first isolated in Europe during the winter of 2017–2018 [51]. Experimental studies have shown that H5N6 has an infectious period from 7 to 10 days in ducks [48,52]. One study found that some strains of H5N6 were highly pathogenic in both chickens and ducks [53]. Though the number of studies is few, this evidence indicates that H5N6 also has a shorter infectious period and higher mortality rate than H5N8, which, consistent with our findings here, may underlie its more limited translocation.

In the early phases following discovery of a novel subtype or strain, targeted laboratory experiments and field surveillance can help assess the risk presented by the new virus. The findings related here suggest conducting experiments to determine the infectivity and transmissibility of the virus, as well as the virus shedding period, in critical avian hosts (e.g. migratory ducks). If the virus is found to possess high transmissibility, a low infection-induced mortality rate and, in particular, a low infection recovery rate (a longer infectious period), more resources should be deployed to monitor and control the spread of the virus and preventive actions will be needed more broadly across the world.

Our study has a number of limitations and areas for future exploration. Our model only simulated the translocation of virus within one homogeneous migratory bird population; however, interactions can occur among different populations and species at staging sites and breeding grounds during migration. Such mixing could allow reintroduction of a virus into a bird population; alternatively,

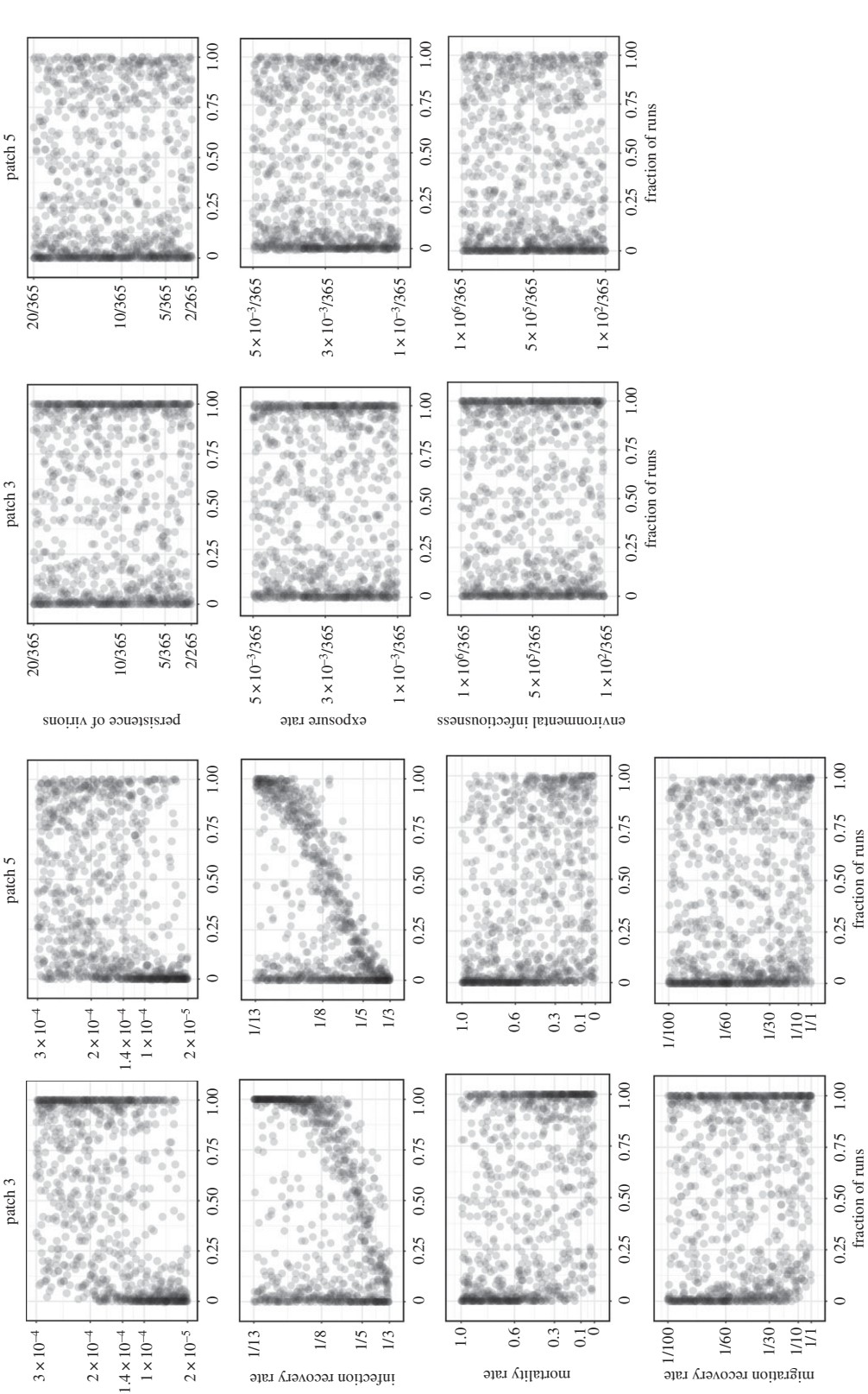

**Figure 4.** Marginal distribution of the likelihood of virus reaching patches 3 and 5 based on the results with the second version of the model including environmental transmission. As in figure 2, every point in the figure represents the result calculated for one sampled parameter combination.

**Table 4.** PRCC between the likelihood of virus translocating to patch 3 or patch 5 and each varying parameter based on the results of the second model with environmental transmission.

| input parameter | patch 3 | | patch 5 | |
|---|---|---|---|---|
| | PRCC | *p*-value | PRCC | *p*-value |
| $\beta$ | 0.6733 | 0.00 | 0.6178 | 0.00 |
| $r$ | −0.6567 | 0.00 | −0.7081 | 0.00 |
| $\mu_d$ | −0.6314 | 0.00 | −0.5695 | 0.00 |
| $v$ | 0.3007 | 0.00 | 0.2943 | 0.00 |
| $C$ | −0.0263 | 0.41 | 0.0484 | 0.12 |
| $\gamma$ | 0.0293 | 0.36 | 0.0102 | 0.75 |
| $\alpha$ | 0.0171 | 0.59 | 0.0259 | 0.41 |

the virus could be translocated by multiple populations [35]. We also used the likelihood of translocation to the summer breeding grounds as a proxy for intercontinental dissemination risk. Dispersal of virus to breeding grounds, such as Alaska, would provide greater opportunity for infection of other birds from other migratory pathways. However, we did not consider bird population interactions at the breeding grounds nor autumn migration to new areas. Future studies could explore these interactions and the ecological and pathobiological factors supporting intercontinental dispersal.

In the study of Breban *et al.* [40], virus rapidly went extinct in an SIR model with pulsed reproduction but no environmental transmission. The authors found that even small levels of environmental transmission could support extended viral persistence in a population and locality. By contrast, our study shows that environmental transmission has little impact on the translocation of infection within a single flock. In our model, the bird population stays at each of the eight patches for only a short duration. After migrating to the next patch, the birds leave the infected habitats, which isolates the birds from the virus in the environment. We therefore conclude that environmental transmission is important for virus persistence in one location, but it does not appear critical for initial translocation in this single flock model. Future work with model systems simulating multiple flocks is needed to examine whether environmental transmission facilitates translocation depending on rates of flock overlap in space and time.

Pathogenicity also differs among young and adult birds [54]. Young birds need more time to recover and mortality rates in young birds are often higher than in adults, which may influence epidemic dynamics and infection dispersal. As a consequence, a model that includes these population dynamics might provide further discrimination.

We also did not consider immunity loss or cross-immunity in the model. It has been observed that immunity in birds wanes following recovery [55], and some studies have shown that LPAI antibodies reduce the probability of developing HPAI infection [54,56,57]. Including these effects in the future would require some strong assumptions regarding process, but could provide insight into how immunity might affect virus transmission and dispersal.

Despite these limitations, our model results indicate that the geographical dispersal of avian influenza is particularly sensitive to the length of the infectious period, and that viruses with a longer infectious period (i.e. a lower recovery rate) are more likely to disperse rapidly. These findings present a working hypothesis that could be tested in both the field and laboratory.

Data accessibility. Data available from the Dryad Digital Repository: https://doi.org/10.5061/dryad.rt552kr [58].

Authors' contributions. X.L. and J.S. designed the study, carried out the analysis; X.L. performed the simulations; X.L., B.X. and J.S. drafted the manuscript. All authors gave final approval for publication.

Competing interests. Jeffrey Shaman and Columbia University disclose partial ownership of SK Analytics.

Funding. This study was supported by Chinese Scholarship Council (no. 201706210329), the Ministry of Science and Technology, China, National Key Research and Development Program (2016YFA0600104) and the US National Institutes of Health (General Medical Sciences) (GM110748).

Acknowledgements. We are grateful to two anonymous reviewers and the editor, who provided comments and substantially improved the manuscript.

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
