## [Reviewer comments · Royal Society Open Science]

Review History

RSOS-190276.R0 (Original submission)

Review form: Reviewer 1

Is the manuscript scientifically sound in its present form?

No

Are the interpretations and conclusions justified by the results?

Yes

Is the language acceptable?

Yes

Is it clear how to access all supporting data?

No

Do you have any ethical concerns with this paper?

No

Have you any concerns about statistical analyses in this paper?

No

Recommendation?

Major revision is needed (please make suggestions in comments)

Comments to the Author(s)

This is an interesting and well-written paper that makes a nice contribution to the study of avian influenza virus in wild birds. The authors numerically solve a compartmental model of AIV in a migratory population, showing primarily that lower recovery rates increase the likelihood of AIV translocation within the migratory population. My major and minor concerns are outlined below:

Major concerns:

1. The primary results rest on a series of model simulations and sensitivity analyses, but it is unclear how the authors varied the four parameters (β , r , μ , ν) and how they quantified sensitivity. From the text (L173-188), I was expecting some kind of Latin hypercube sampling approach to vary most possible combinations of parameters, but Figure 3 suggests that only four a priori values per parameter were assessed. Did you simply assess all possible combinations of these four values across the four parameters? It is also unclear how "sensitivity" of results to parameter variation was quantified (e.g., L211), as no standard measures of sensitivity (e.g., partial rank correlations) are included. I would encourage either justifying the use of these four values per parameter or using something like the LHS approach to better sample parameter space and measure sensitivity of your model outcomes to parameter variation.
2. While the key results of the model analysis may be novel within the AIV system (L251-260), I think it's worth noting in your discussion that many of these insights could be generated from a far simpler SIR model (e.g., a simple SI/SIS/SIR would also tell you that a lower infection recovery rate will increase transmission potential). Some clarification of what this model tells us beyond a more parsimonious model could help improve clarity.
3. A very helpful contribution of your analysis is to guide field surveillance and control (L285-288). Because you note that these pathobiological features could help prioritize surveillance and control, you should also discuss how researchers could glean this information about a particular AIV strain. For example, how could one assess a high transmission rate or a high recovery rate in the early phases of strain discovery?
4. While the paper has no empirical data, I strongly suggest the authors upload their R code to increase reproducibility.

Minor comments:

L71: This statement on concerns for human pandemics could use a citation

L132: change "recovery" to "recover"

L209: It would help to define marginal likelihood and how this was quantified

L217 and elsewhere: It may be more intuitive to denote recovery rate as the duration of the infection period (e.g., 13 days) rather than the current form (recovery rate of 1/13 day⁻¹).

L226: I was a little confused about the use of density histograms. Wouldn't these just show you

the distribution of parameters you originally used in your simulations?

Review form: Reviewer 2

Is the manuscript scientifically sound in its present form?

No

Are the interpretations and conclusions justified by the results?

Yes

Is the language acceptable?

Yes

Is it clear how to access all supporting data?

No

Do you have any ethical concerns with this paper?

No

Have you any concerns about statistical analyses in this paper?

No

Recommendation?

Major revision is needed (please make suggestions in comments)

Comments to the Author(s)

The manuscript describes a modeling study that investigates the role of various infection parameters in the spatial spread of influenza virus. I find the study and approach interesting, but have a major concern about the model that I believe needs to be addressed. Therefore, I recommend major revision.

Major concerns:

1. It's not entirely clear that this will change the conclusions reached by the study, but I believe that the interplay between illness and migration has not been correctly formulated. The model assumes that infected birds in I1 and recovering birds in R1 are too sick to migrate, but that birds in I2 (which comes AFTER I1) and R2 can migrate. This basically suggests that immediately upon infection, birds are too ill to fly, but as the disease progresses, they are again able to fly. However, once they start to recover they are again too ill to migrate, but then finally recover enough to migrate again. It would make more sense to me if I2 and R1 were the two groups that are unable to migrate.
2. The authors state that the simulations are stochastic, but do not describe the algorithm used to solve the model equations stochastically. Additionally, their code does not seem to be available anywhere.

Minor comments:

1. The model includes 9 patches, but the analyses only considers transmission of virus up to patch 5. It's not clear to me why the authors are including the remaining patches if they don't analyze what happens when the birds move there. In fact, the authors added the 9th patch from a previous 8 patch model, which seems like a useless addition since it is never used.

2. Figures 2 and 4 are very difficult to interpret. There needs to be more guidance in the manuscript on what exactly is being plotted.

Decision letter (RSOS-190276.R0)

07-Mar-2019

Dear Ms Li,

The editors assigned to your paper ("Pathobiological features favoring the intercontinental dissemination of highly pathogenic avian influenza virus") have now received comments from reviewers. We would like you to revise your paper in accordance with the referee and Associate Editor suggestions which can be found below (not including confidential reports to the Editor). Please note this decision does not guarantee eventual acceptance.

Please submit a copy of your revised paper before 30-Mar-2019. Please note that the revision deadline will expire at 00.00am on this date. If we do not hear from you within this time then it will be assumed that the paper has been withdrawn. In exceptional circumstances, extensions may be possible if agreed with the Editorial Office in advance. We do not allow multiple rounds of revision so we urge you to make every effort to fully address all of the comments at this stage. If deemed necessary by the Editors, your manuscript will be sent back to one or more of the original reviewers for assessment. If the original reviewers are not available, we may invite new reviewers.

- Data accessibility

It is a condition of publication that all supporting data are made available either as supplementary information or preferably in a suitable permanent repository. The data accessibility section should state where the article's supporting data can be accessed. This section

should also include details, where possible of where to access other relevant research materials such as statistical tools, protocols, software etc can be accessed. If the data have been deposited in an external repository this section should list the database, accession number and link to the DOI for all data from the article that have been made publicly available. Data sets that have been deposited in an external repository and have a DOI should also be appropriately cited in the manuscript and included in the reference list.

If you wish to submit your supporting data or code to Dryad (<http://datadryad.org/>), or modify your current submission to dryad, please use the following link:
<http://datadryad.org/submit?journalID=RSOS&manu=RSOS-190276>

- **Competing interests**

- **Authors' contributions**

- **Acknowledgements**

- **Funding statement**

on behalf of Prof Kevin Padian (Subject Editor)
openscience@royalsociety.org

Associate Editor's comments :

Associate Editor: 1

Comments to the Author:

Two reviewers have commented on your paper, and, while each find merit in the work, they recommend substantial changes to the manuscript. Good luck and we will look forward to receiving your revision in the near future.

Comments to Author:

Reviewers' Comments to Author:

Reviewer: 1

Comments to the Author(s)

This is an interesting and well-written paper that makes a nice contribution to the study of avian influenza virus in wild birds. The authors numerically solve a compartmental model of AIV in a migratory population, showing primarily that lower recovery rates increase the likelihood of AIV translocation within the migratory population. My major and minor concerns are outlined below:

Major concerns:

1. The primary results rest on a series of model simulations and sensitivity analyses, but it is unclear how the authors varied the four parameters (β , r , μ , ν) and how they quantified sensitivity. From the text (L173-188), I was expecting some kind of Latin hypercube sampling approach to vary most possible combinations of parameters, but Figure 3 suggests that only four a priori values per parameter were assessed. Did you simply assess all possible combinations of these four values across the four parameters? It is also unclear how "sensitivity" of results to parameter variation was quantified (e.g., L211), as no standard measures of sensitivity (e.g., partial rank correlations) are included. I would encourage either justifying the use of these four values per parameter or using something like the LHS approach to better sample parameter space and measure sensitivity of your model outcomes to parameter variation.
2. While the key results of the model analysis may be novel within the AIV system (L251-260), I think it's worth noting in your discussion that many of these insights could be generated from a far simpler SIR model (e.g., a simple SI/SIS/SIR would also tell you that a lower infection recovery rate will increase transmission potential). Some clarification of what this model tells us beyond a more parsimonious model could help improve clarity.
3. A very helpful contribution of your analysis is to guide field surveillance and control (L285-288). Because you note that these pathobiological features could help prioritize surveillance and control, you should also discuss how researchers could glean this information about a particular AIV strain. For example, how could one assess a high transmission rate or a high recovery rate in the early phases of strain discovery?
4. While the paper has no empirical data, I strongly suggest the authors upload their R code to increase reproducibility.

Minor comments:

L71: This statement on concerns for human pandemics could use a citation

L132: change “recovery” to “recover”

L209: It would help to define marginal likelihood and how this was quantified

L217 and elsewhere: It may be more intuitive to denote recovery rate as the duration of the infection period (e.g., 13 days) rather than the current form (recovery rate of 1/13 day⁻¹).

L226: I was a little confused about the use of density histograms. Wouldn't these just show you the distribution of parameters you originally used in your simulations?

Reviewer: 2

Comments to the Author(s)

The manuscript describes a modeling study that investigates the role of various infection parameters in the spatial spread of influenza virus. I find the study and approach interesting, but have a major concern about the model that I believe needs to be addressed. Therefore, I recommend major revision.

Major concerns:

1. It's not entirely clear that this will change the conclusions reached by the study, but I believe that the interplay between illness and migration has not been correctly formulated. The model assumes that infected birds in I1 and recovering birds in R1 are too sick to migrate, but that birds in I2 (which comes AFTER I1) and R2 can migrate. This basically suggests that immediately upon infection, birds are too ill to fly, but as the disease progresses, they are again able to fly. However, once they start to recover they are again too ill to migrate, but then finally recover enough to migrate again. It would make more sense to me if I2 and R1 were the two groups that are unable to migrate.
2. The authors state that the simulations are stochastic, but do not describe the algorithm used to solve the model equations stochastically. Additionally, their code does not seem to be available anywhere.

Minor comments:

1. The model includes 9 patches, but the analyses only considers transmission of virus up to patch 5. It's not clear to me why the authors are including the remaining patches if they don't analyze what happens when the birds move there. In fact, the authors added the 9th patch from a previous 8 patch model, which seems like a useless addition since it is never used.
2. Figures 2 and 4 are very difficult to interpret. There needs to be more guidance in the manuscript on what exactly is being plotted.

Author's Response to Decision Letter for (RSOS-190276.R0)

See Appendix A.

RSOS-190276.R1 (Revision)

Review form: Reviewer 1

Is the manuscript scientifically sound in its present form?

Yes

Are the interpretations and conclusions justified by the results?

Yes

Is the language acceptable?

Yes

Is it clear how to access all supporting data?

Yes

Do you have any ethical concerns with this paper?

No

Have you any concerns about statistical analyses in this paper?

No

Recommendation?

Accept as is

Comments to the Author(s)

The authors have addressed all my previous concerns.

Review form: Reviewer 2

Is the manuscript scientifically sound in its present form?

Yes

Are the interpretations and conclusions justified by the results?

Yes

Is the language acceptable?

Yes

Is it clear how to access all supporting data?

Yes

Do you have any ethical concerns with this paper?

No

Have you any concerns about statistical analyses in this paper?

No

Recommendation?

Accept as is

Comments to the Author(s)

The authors have addressed all comments to my satisfaction. The paper is much improved and I recommend it be accepted.

Decision letter (RSOS-190276.R1)

08-Apr-2019

Dear Ms Li,

I am pleased to inform you that your manuscript entitled "Pathobiological features favoring the intercontinental dissemination of highly pathogenic avian influenza virus" is now accepted for publication in Royal Society Open Science.

on behalf of Professor Kevin Padian (Subject Editor)
openscience@royalsociety.org

Associate Editor Comments to Author:

The reviewers consider your manuscript to be ready for acceptance - congratulations! The editorial team and production team will be in touch with advice on how to proceed. Thank you for considering our journal for this submission.

Reviewer comments to Author:

Reviewer: 1

Comments to the Author(s)

The authors have addressed all my previous concerns.

Reviewer: 2

Comments to the Author(s)

The authors have addressed all comments to my satisfaction. The paper is much improved and I recommend it be accepted.

Appendix A

Response to Reviewer 1's comments

We thank the reviewer for his/her helpful comments and suggestions. They are incorporated into the revision of the manuscript. Below is a point-by-point response to the reviewer's specific comments. The original comments are in italics and our response is in normal font.

Comments to the Author(s)

This is an interesting and well-written paper that makes a nice contribution to the study of avian influenza virus in wild birds. The authors numerically solve a compartmental model of AIV in a migratory population, showing primarily that lower recovery rates increase the likelihood of AIV translocation within the migratory population. My major and minor concerns are outlined below:

We thank the reviewer for the positive comments.

Major concerns:

- 1. The primary results rest on a series of model simulations and sensitivity analyses, but it is unclear how the authors varied the four parameters (β , r , μ , ν) and how they quantified sensitivity. From the text (L173-188), I was expecting some kind of Latin hypercube sampling approach to vary most possible combinations of parameters, but Figure 3 suggests that only four a priori values per parameter were assessed. Did you simply assess all possible combinations of these four values across the four parameters? It is also unclear how "sensitivity" of results to parameter variation was quantified (e.g., L211), as no standard measures of sensitivity (e.g., partial rank correlations) are included. I would encourage either justifying the use of these four values per parameter or using something like the LHS approach to better sample parameter space and measure sensitivity of your model outcomes to parameter variation.*

After reading the reviewer's comments, we re-ran the baseline model with 600 parameter combinations sampled using LHS. To quantify the sensitivity and present these findings in the revised manuscript, we have calculated partial rank correlation coefficients based on the results from re-running the model. We have re-drawn Figures 2, 3 and 4, and the results of the partial rank correlation are shown in Table 3 and Table 4. Model findings remain unchanged.

2. *While the key results of the model analysis may be novel within the AIV system (L251-260), I think it's worth noting in your discussion that many of these insights could be generated from a far simpler SIR model (e.g., a simple SI/SIS/SIR would also tell you that a lower infection recovery rate will increase transmission potential). Some clarification of what this model tells us beyond a more parsimonious model could help improve clarity.*

Thank you for your comments and suggestion. In this model, we considered bird movement and virus translocation along the birds' flyway, in order to simulate AIV dissemination. Compared to a simpler SIR model, this model form not only considers three infection states for the birds—susceptible, infected, and recovered—but also loss and recovery of the ability to migrate induced by infection. These latter conditions can impact the dynamic of infection translocation along the entire flyway. According to Galsworthy et al. (2011), after including migration delay, the scale of an outbreak becomes smaller than in a simpler SIR model without migration delay, which is attributed to infected birds being isolated from the main population of susceptible birds. We have added some clarification on this issue to the revised manuscript (L272 – L281).

3. *A very helpful contribution of your analysis is to guide field surveillance and control (L285-288). Because you note that these pathobiological features could help prioritize surveillance and control, you should also discuss how researchers could glean this information about a particular AIV strain. For example, how could one assess a high transmission rate or a high recovery rate in the early phases of strain discovery?*

Thank you for your suggestion. We have added discussion of these issues to the revised manuscript that illustrate what information might be provided for AIV laboratory experiments and how researchers might assess the risk of a novel virus (L315– L322).

4. *While the paper has no empirical data, I strongly suggest the authors upload their R code to increase reproducibility.*

We have uploaded the R code to the Dryad Digital Repository:

Minor comments:

- *L71: This statement on concerns for human pandemics could use a citation*

Citations for the statement on concerns for human pandemics have been added (L71).

- *L132: change “recovery” to “recover”*

We have corrected the wording.

- *L209: It would help to define marginal likelihood and how this was quantified*

Thank you for your suggestion. We have added some explanation of marginal likelihood in the revised manuscript (L211 to L214).

- *L217 and elsewhere: It may be more intuitive to denote recovery rate as the duration of the infection period (e.g., 13 days) rather than the current form (recovery rate of 1/13 day⁻¹).*

We have fixed the expression following your suggestion (L220 – L226).

- *L226: I was a little confused about the use of density histograms. Wouldn't these just show you the distribution of parameters you originally used in your simulations?*

Thank you for this question. The density histograms do not just show the distribution of parameters originally used in the simulations. They show the distribution of parameters that support AIV always translocating to either Patch 3 or Patch 5, as well as never translocating to Patch 3 or Patch 5. Based on the result of this analysis, we identify the parameter combinations that always support virus reaching a given patch (i.e. 100% of the 100 stochastic simulations with a parameter

combination). We also recorded the parameter combinations for which virus never reaches a given patch (i.e. 0% of the stochastic runs). We then plotted the marginal density histograms of always supporting translocation and never supporting translocation for each parameter.

Response to Reviewer 2's comments

We thank the reviewer for his/her constructive and helpful comments. Many of these comments are incorporated into the revision of the manuscript. Below we provide a point-by-point response. The reviewer's original comments are in italics and our response is in normal font.

Comments to the Author(s)

The manuscript describes a modeling study that investigates the role of various infection parameters in the spatial spread of influenza virus. I find the study and approach interesting, but have a major concern about the model that I believe needs to be addressed. Therefore, I recommend major revision.

We thank the reviewer for the positive comments.

Major concerns:

- 1. It's not entirely clear that this will change the conclusions reached by the study, but I believe that the interplay between illness and migration has not been correctly formulated. The model assumes that infected birds in I1 and recovering birds in R1 are too sick to migrate, but that birds in I2 (which comes AFTER I1) and R2 can migrate. This basically suggests that immediately upon infection, birds are too ill to fly, but as the disease progresses, they are again able to fly. However, once they start to recover, they are again too ill to migrate, but then finally recover enough to migrate again. It would make more sense to me if I2 and R1 were the two groups that are unable to migrate.*

Thank you for the comments and suggestions. In this model, once the birds are infected (birds in I1), they are unable to migrate. Infected birds in I1 can recover but remain unable to migrate, R1, then subsequently regain their migratory ability to R2, or infected birds in I1 can regain the migratory ability while remaining infectious, I2, then subsequently recover to R2. That is, the infection recovery and migration recovery are independent of each other, and the birds can first recover from infection then regain the ability to migrate, or first regain the ability to migrate then recover from infection. The flowchart of the model is shown in Figure R2.1. As the reviewer mentioned that birds might still fly before developing a full infection, we tested the addition of a

latent period before I1 to the model as well. The marginal likelihood of reaching Patch 3 and Patch 5 is shown in Figure R2.2. The differences between Figure R2.2 and Figure 2 in the old version of the manuscript are nominal, indicating that addition of a pre-infection latent period to the model does not change the sensitivity of virus translocation to the four epidemiological features of interest.

Figure R2.1 Model structure showing the movement of birds between compartments within Patch i . The five compartments are: Susceptible (S), infected and unable to migrate (I1), infected and able to migrate (I2), recovered and unable to migrate (R1), or recovered and able to migrate (R2). Once infected, birds move from S to I1 and lose the ability to migrate. Birds regain the ability to migrate at rate v , and recover from infection at rate r . Migration can only happen for birds in compartments S, I2, and R2.

Figure R2.2 Marginal distribution of the likelihood of virus reaching Patches 3 and 5 based on the results with the model including a latent period. The x axis represents the likelihood of virus reaching Patches 3 and 5. The likelihood was quantified as the fraction of runs among 100 stochastic simulations for which virus reached Patch 3 or Patch 5. The size of the circle represents the number of points overlapping at the same location in the figure.

2. The authors state that the simulations are stochastic, but do not describe the algorithm used to solve the model equations stochastically. Additionally, their code does not seem to be available anywhere.

To conduct simulations, the model equations were used to determine transition expectance based on the current system state and parameters. A random Poisson process was then used to select the actual rates of state transition using this mean expectance. Additionally, we have uploaded the R code to the Dryad Digital Repository:

<https://datadryad.org/review?doi=doi:10.5061/dryad.rt552kr>

Minor comments:

- 1. The model includes 9 patches, but the analysis only considers transmission of virus up to patch 5. It's not clear to me why the authors are including the remaining patches if they don't analyze what happens when the birds move there. In fact, the authors added the 9th patch from a previous 8 patch model, which seems like a useless addition since it is never used.*

The reviewer is correct. We now just describe 8 patches (L114 – L117). Only translocation to patch 5 is studied, as we are interested in the initial dispersal of the virus (not its return back where it is already present). The importance of patch 5 is discussed at the beginning of the Results section.

- 2. Figures 2 and 4 are very difficult to interpret. There needs to be more guidance in the manuscript on what exactly is being plotted.*

Thank you for your comment and suggestion. Based on Reviewer 1's comments, we re-ran the model with sampled points using a Latin hypercube sampling (LHS) approach. Using these new results, we have re-drawn Figures 2 and 4. We have also added more description for Figures 2 and 4 in the Result section (L210 - L213; L255 - L257) and in the captions of the figures (L588 – L593; L604 – L606).